# Ultrasound-Assisted Extraction of Antioxidants from *Melastoma malabathricum* Linn.: Modeling and Optimization Using Box–Behnken Design

**DOI:** 10.3390/molecules28020487

**Published:** 2023-01-04

**Authors:** Suzziyana Hosni, Siti Salwa Abd Gani, Valérie Orsat, Masriana Hassan, Sumaiyah Abdullah

**Affiliations:** 1Halal Products Research Institute, Universiti Putra Malaysia, Seri Kembangan, Serdang 43400, Selangor, Malaysia; 2Department of Agriculture Technology, Faculty of Agriculture, Universiti Putra Malaysia, Seri Kembangan, Serdang 43400, Selangor, Malaysia; 3Natural Medicine and Products Research Laboratory, Institute of Biosceince, Universiti Putra Malaysia, Seri Kembangan, Serdang 43400, Selangor, Malaysia; 4Macdonald Campus, McGill University, Lakeshore Road, Sainte-Anne-de-Bellevue, QC 21111, Canada; 5Department of Pathology, Faculty of Medicine and Health Sciences, Universiti Putra Malaysia, Seri Kembangan, Serdang 43400, Selangor, Malaysia; 6Department of Plant Protection, Faculty of Agriculture, Universiti Putra Malaysia, Seri Kembangan, Serdang 43400, Selangor, Malaysia

**Keywords:** *Melastoma malabathricum*, ultrasound-assisted extraction (UAE), one-factor-at-a-time (OFAT), response surface methodology (RSM), Box–Behnken design (BBD), antioxidant assay, physicochemical characterization, phytochemical screening

## Abstract

This study presents modeling and optimization of ultrasound-assisted extraction (UAE) of *Melastoma malabathricum* with the objective of evaluating its phytochemical properties. This one-factor-at-a-time (OFAT) procedure was conducted to screen for optimization variables whose domains included extraction temperature (X_ET_), ultrasonic time (X_UT_), solvent concentration (X_SC_), and sample-to-liquid ratio (X_SLR_). Response surface methodology (RSM) coupled with Box–Behnken design (BBD) was applied to establish optimum conditions for maximum antioxidant extraction. Modeling and optimization conditions of UAE at 37 kHz, X_ET_ 32 °C for X_UT_ 16 min and dissolved in an X_SC_ 70% ethanol concentration at a X_SLR_ 1:10 ratio yielded scavenging effects on 2,2-diphenyl-1-picryl-hydrazyl (DPPH) at 96% ± 1.48 and recorded values of total phenolic content (TPC) and total flavonoid content (TFC) at 803.456 ± 32.48 mg GAE (gallic acid equivalents)/g, and 102.972 ± 2.51 mg QE (quercetin equivalents)/g, respectively. The presence of high flavonoid compounds was verified using TWIMS-QTOFMS. Chromatic evaluation of phytochemicals using gas chromatography–mass spectrometry (GC–MS) revealed the presence of 14 phytocompounds widely documented to play significant roles in human health. This study provides a comparative evaluation with other studies and may be used for validation of the species’ potential for its much-acclaimed medicinal and cosmeceutical uses.

## 1. Introduction

*Melastoma malabathricum*, also known by various vernacular names depending on its geographical location, is one of the evergreen shrubs in the family *Melastomaceae* and one of 12 species commonly found in Malaysia. Several species in the family have been comprehensively documented to have imperative biological roles in human health for their widely acknowledged therapeutic values. The whole plant parts of *M. malabathricum,* including leaves, flowers, and fruits (Figure 1), have been reported to have been traditionally used in treatments of various illnesses [1,2,3].

Despite decades of research on, and acknowledgement of, the importance of the species as a medicinal herb, there is still much to be investigated. The literature has it that documented clinical data on the species’ efficacies in the treatments of some illnesses are, at best, very limited or have not been sufficiently far-reaching and pharmacologically attested. This situation compels the need for further empirical studies with a view of establishing its potential in treatments of numerous illnesses. The primary objective of this study was to optimize extraction of its bioactive compounds using ultrasound-assisted extraction (UAE) procedures.

Ultrasonic-assisted extraction (UAE) helps to improve the extraction process at low temperatures while inflicting minimal damage to the structural and molecular characteristics of the chemicals in plant materials. Due to its advantages over conventional extraction procedures, including shorter time, less use of solvents, a higher extraction yield, and a lower operating cost, UAE has commanded a substantial increase in interest [4,5].

The extraction optimization methodology has enabled the present study to develop an appropriate extraction process with a time-efficient execution of experiments. The conditions for extraction of bioactive compounds were optimized, and determined the antioxidant properties using an empirical or statistical method. Box–Behnken designs coupled with response surface methodology (RSM) models were applied to determine the optimal extraction conditions. The optimization procedure based on RSM is an efficient tool to forecast the conditions leading to optimal responses. RSM’s Box–Behnken design (BBD) is specifically made to match a second-order model, the focus of most RSM studies. The BBD requires three levels of each factor to fit a second-order regression model (quadratic model). A one-factor-at-a-time (OFAT) procedure was conducted for screening the importance of each parameter and to find the range for each selected parameter. Several protocols have been reported for the extraction of plants’ bioactive components using different conditions for extraction such as different reaction times, solvents, solvent concentrations, pH, and different compounds used as antioxidant standards.

The study was undertaken with the objective of optimizing conditions for a high yield of antioxidant activity through inhibition of 2,2-di(4-tert-octylphenyl)-1-picrylhydrazyl (DPPH), total phenolic content (TPC), and total flavonoid content (TFC) of plant extracts from *Melastoma malabathricum* leaves. The optimized extract was then evaluated for its phytochemicals as well as physicochemical properties using gas chromatography–mass spectrometry (GC–MS) analysis, a sophisticated analytical technique needed for complex metabolomic investigations to support research into primary and secondary metabolites in plants and to obtain comprehensive coverage of compounds, as well as quadrupole time-of-flight mass spectrometry (QTOF-MS) analysis, the recent advanced LC–QTOF-MS technology which offers improved resolution and mass precision for native mass spectrometry analysis, biopharmaceutical characterization, and intact protein analysis.

## 2. Materials and Methods

### 2.1. Materials

#### 2.1.1. Chemicals and Reagents

All chemicals and reagents used in the experiments were of analytical grade (Sigma–Aldrich (M) Sdn. Bhd. Subang Jaya, an affiliate of Merck KGaA, Darmstadt, Germany) and purchased from local suppliers/importers. The chemicals included methanol, absolute ethanol, the Folin–Ciocalteu reagent, gallic acid, aluminium trichloride, as well as a 1,1-diphenyl-2-picrylhydrazyl radical (DPPH).

#### 2.1.2. Authentication of Species

Authentication of *M. malabathricum* was performed by the Institute of Bioscience (IBS), Universiti Putra Malaysia (UPM), Serdang, Selangor, Malaysia. A voucher (UPM IBS/UB/H37/21) of herbarium specimens for the plant samples was deposited at IBS.

### 2.2. Methods

#### 2.2.1. Plant Material and Sample Preparation

Leaves of *M. malabathricum* were collected from mature plants growing freely at the periphery of a farm area on the UPM campus. Leaves were utilized in the present study following their wide use in numerous pharmacological studies on the species [6,7,8]. Harvested leaf samples were cleaned and washed using tap water and subsequently rinsed with distilled water. Leaf samples were oven-dried at 50 ± 2 °C for 2 days following the method by [9] with minor modifications. Leaf samples of about 100 g each were weighed, ground into a fine powder to an approximately uniform grain or fragment size of 1 mm using a mechanical grinder, and placed in separate airtight containers in a 4 °C chiller before subsequent extraction procedures were performed.

#### 2.2.2. Single Factors for Extraction Procedures

In the present study, initial single-factor experiments were conducted using a one-factor-at-a time (OFAT) method to determine the experimental domain and the best range of conditions for an appropriate RSM design to be used for UAE. Four parameters, namely extraction temperature (X_ET_), ultrasonic time (X_UT_), solvent concentration (X_SC_), and sample-to-liquid ratio (X_SLR_), were selected as the response for this design.

To select the independent variables, the single factors for the extraction procedures were set as follows. The influence of X_UT_ (20, 50, 80 min) on the antioxidant content was first determined under the following fixed conditions: ultrasonic frequency of 37 Hz, X_SLR_ (1:20 g/mL) and X_ET_ 55 °C. Secondly, the impact of X_SLR_ (1:10, 1:50, 1:80) on the antioxidant content was determined under the following fixed conditions: ultrasonic frequency of 37 Hz, X_UT_ 30 min, X_ET_ 55 °C. Finally, the influence of X_ET_ (30, 60, 90 °C) on antioxidant yield was determined under the following fixed conditions: ultrasonic frequencies of 37 Hz, X_SLR_ (1:20 g/mL), X_UT_ 30 min. Based on OFAT, RSM was used to estimate the optimization conditions for the maximum yield of extraction.

#### 2.2.3. Optimization of Extraction Conditions Using Response Surface Methodology (RSM)

##### Experimental Design

Software Design Expert 7.0.0 (Trial version, Stat-Ease Inc., Minneapolis, MN, USA) was used for the application of the Box–Behnken design with RSM for modeling and optimization of extraction. Four independent variables were considered: X_ET_, X_UT_, X_SC_ and X_SLR_. Power calculations were evaluated over −1 to +1 coded factor spaces as shown in Table 1.

Following the Box–Behnken design, the range of values for each independent variable were as follows: X_ET_: 30–70 °C, X_UT:_ 10–60 min, X_SC:_ 20–70%, and X_SLR:_ 1:10–1:50. Using this design, the number of experimental sets was reduced without affecting the accuracy of optimization compared with traditional factorial design methods. All uncoded factors had their own units. By introducing coded variables, the factors were limitless. The effects of these four variables on antioxidant activities could be predicted using a second-order polynomial model, as shown in Equation (1):y = b_0_ + b_1_x_1_ + b_2_x_2_ + b_3_x_3_ + b_4_x_4_ + b_12_x_1_x_2_ + b_13_x_1_x_3_ + b_14_x_1_x_4_+ b_23_x_2_x_3_+ b_24_x_2_x_4_ + b_34_x_3_x_4_ +b_11_x_1_ + b_22_x_2_ + b_33_x_3_ + b_44_x_4_(1)
where y = the predicted response; b_0_ = constant; b_1_, b_2_, b_3_, b_4_ = regression coefficients for linear effects; b_11_, b_22_, b_33_, b_44_ = quadratic coefficients; b_12_, b_13_, b_14_, b_23_, b_24_, b_34_ = interaction coefficients; and x_1_, x_2_, x_3_, x_4_ = parameters considered.

##### Ultrasound-Assisted Extraction (UAE)

Extraction of antioxidant compounds was conducted to separate and characterize the constituents of the species under study. Extraction was performed using the ultrasound-assisted extraction method (UAE) using a sonic generator (Model Elmasonic 8–30 H) fitted with an ultrasound probe (Figure 2). Erlenmeyer flasks containing the extracts were submerged in water up to the level of extracts. Prior to extraction, optimization of extraction conditions to maximize the efficiency of extraction in the recovery of compounds of interest was undertaken.

While the ultrasonic frequency was set at 37 kHz, the X_ET_, X_UT_, X_SC_ and X_SLR_ were established as the independent variables. The Box–Behnken design (BBD) coupled with RSM was used to determine the optimal conditions for extraction, generating a total of 29 conditions as presented in Table 2. Each experiment was carried out in triplicate. Narrow-neck Erlenmeyer flasks were used for heating to minimize content loss through evaporation. The extract was then removed from the sonicator and filtered before a rotary evaporator (rotavap) procedure was performed to remove the sample’s solvent through evaporation and to collect the crude extract. The crudes were then stored in amber glass bottles and placed in a chiller set at 4 °C prior to subsequent procedures for preliminary phytochemical assessment and evaluation of antioxidant capacity.

##### Determination of Antioxidants in *M. malabathricum* Leaf Extracts

In reference to [10], it is not possible to determine the antioxidant capacity of plant extracts using just one test method. Additionally, the experimental setup and test principle of each antioxidant test differs. Because the protocols and experimental settings for various techniques differ, many antioxidants are used as controls for various test methods based on their purification rates and duration. Antioxidants can neutralize free radicals by either donating hydrogen or providing electrons, and they can also be polar or non-polar. Consequently, various antioxidant tests utilize a variety of control antioxidants. As a result, three antioxidant tests were used in this study to determine the best extraction conditions for *M. malabathricum* leaves and plant leaves in general.

##### Free Radical Scavenging Activity (DPPH)

The antioxidant activity of extracts of *M. malabathricum* was determined by measuring free-radical-scavenging activity using a 1,1-diphenyl-2-picrylhydrazyl (DPPH) method as proposed by [10]. In the procedure, 10 mg of DPPH was dissolved in 100 mL of ethanol. Then 150 µL of ethanolic solution of DPPH was transferred to a 96-well microliter plate and mixed with 50 µL of the extract sample. Each sample was analyzed in triplicate. Ascorbic acid was used as the positive control, while ethanol was used as the negative control (blank). The mixture was incubated for 30 min in a dark room prior to measuring using a UV-VIS Microplate Reader (Spectra Max Plus 384, Molecular Devices Co. Ltd., Derwood, MD, USA). The absorbance was recorded at a wavelength of 517 nm. The percentage inhibition of radical scavenging activities of the extract was calculated using the following equation:
(2)$$\%\;\mathrm{Radical}\mathrm{scavenging}\;=\;\frac{\mathrm{Abs}.\;\mathrm{control}-\mathrm{Abs}.\;\mathrm{sample}\;}{\mathrm{Abs}.\;\mathrm{control}}\;\times \;100\%$$
Abs control: Absorbance of DPPH radical + ethanol
Abs sample: Absorbance of DPPH radical + sample extract/standard

##### Total Phenolic Content (TPC)

The total phenolic content was analyzed using procedures described by [11] with minor modifications. Standard curves were made with gallic acid solutions prepared in a concentration dilution series of 6.25, 12.5, 25, 50, 100, and 200 µg/mL. Then 10 µL of each concentration of the standard solution and sample extract were placed in a 96-well microplate. Each sample was prepared in triplicate. Subsequently, 50 µL of 10% Folin–Ciocalteu solution was added and incubated for 5 min. Next, 40 µL of 7.5% Na_2_CO_3_ was mixed in the solution and incubated for 2 h in the dark at room temperature. The absorbance was measured at a wavelength of 750 nm using a UV-VIS Microplate Reader. Curve standards were made by plotting a graph of concentrations (µg/mL) versus absorbance (nm). The regression equation of a standard curve used was as follows:y = ax + b, R^2^ = c(3)
where x: concentration; y: absorbance; and total phenol was expressed in gallic acid equivalents (GAE) mg per g of dried extract.

##### Total Flavonoid Content (TFC)

The total flavonoid content in each sample was measured following procedures proposed by [12]. An amount of 100 µL (1 mg/mL) of each sample was mixed with 2% of AlCl_3_ and incubated for 15 min at room temperature. The absorbance was measured at 425 nm. The same procedure was repeated for a standard solution of quercetin, and the calibration line was obtained. The concentration of flavonoid was read on a calibration line based on measured absorbance. The content of flavonoids in the extracts was expressed in terms of quercetin equivalent, QE (mg of quercetin/g of extract).

#### 2.2.4. Phytochemical Characterization

##### Gas Chromatography–Mass Spectrometry (GC-MS) Analysis

Sample preparation: Approximately 1.0 g of crude extract was put into a 15 mL polypropylene centrifuge tube before 5 mL of ethanol was added. The ethanolic sample was mixed well using a vortex mixer to thoroughly disperse the entire sample. The sample centrifuged for 10 min at 5000× *g* or greater. Then 0.50 mL of the supernatant was transferred to an autosampler vial and was ready for analysis.

Thermo GC-TRACE Ultra ver. 5.0, Thermo MS DSQ II, a GC–MS instrument from Thermo Scientific Co., was used to conduct phytochemical analysis of the ethanolic extract. The GC–MS system underwent the following experimental conditions: a TR 5-MS capillary standard non-polar column, 30 m in length, 0.25 mm ID, with a 0.25 mm film thickness. The mobile phase flow was set at 1.0 mL/min (carrier gas: He). In the gas chromatography section, the injection volume was 1 µL, and the temperature programmed (oven temperature) was 40 °C, rising to 250 °C at 5 °C/min. Using the Wiley spectral library search tool, the findings of samples that had been thoroughly analyzed at a range of 50–650 *m*/*z* were compared.

##### Quadrupole Time-of-Flight Mass Spectrometry (QTOF-MS) Analysis

First, 1 g of plant extract was weighed in a 50 mL centrifuge tube, and 20 mL of 70% ethanol was added. The mixture was vortexed for 1 min and then shaken using a shaker, SPEX SamplePrep 1500 ShaQer (Metuchen, NJ, USA), for 50 min. The sample was subsequently centrifuged at 12,000 rpm for 5 min at 4 °C. Finally, 1 mL of extract was filtered with a 0.2 µm PVDF syringe filter (Agilent Technologies, Santa Clara, CA, USA) and dispensed into a 1.5 mL vial before being injected into an LC-QTOF-MS. Ultra-high performance liquid chromatography (UHPLC) was performed using an ACQUITY UPLC I-Class system from Waters (Manchester, UK). Phenolic compounds were chromatographically separated using an ACQUITY UPLC HSS T3 (100 mm × 2.1 mm × 1.8 μm) column, also from Waters (Manchester, UK), maintained at 40 °C.

A linear binary gradient of water (0.1% formic acid) and acetonitrile (Mobile phase B) were used as Mobile Phase A and B, respectively. The mobile phase composition was changed during the run as follows: 0 min, 1% B; 0.5 min, 1% B; 16.00 min, 35% B; 18.00 min, 100% B; 20.00 min, 1% B. The flow rate was set to 0.6 mL/min, and the injection volume was 1 μL. The UHPLC system was coupled with a Vion IMS QTOF hybrid mass spectrometer from Waters (Manchester, UK). The ion source was operated in positive and negative electrospray ionization (ESI) mode under the conditions set in Table 3:

Nitrogen (>99.5%) was employed as a desolvation and cone gas. Data were acquired in high-definition MS^E^ (HDMS^E^) mode in the range of *m*/*z* 50–1500 at 0.1 s/scan. Argon (99.999%) was used as a collision-induced dissociation (CID) gas.

#### 2.2.5. Statistical Analysis

Data were analyzed using analysis of variance (ANOVA) to determine the lack of fit and the effects of linear, quadratic, and interaction variables on all responses. Data analyses and RSM were performed with Design Expert software (Version 8; Stat-Ease, Inc., Minneapolis, MN, USA).

## 3. Results and Discussion

### 3.1. Sampling

Leaves of *M. malabathricum* were used in the present study. This species has been widely documented to possess high bioactive compounds. There exists an extensive amount of literature on the traditional use of the species which claims to have various medicinal values. Several researchers have compiled up-to-date, extensive, and comprehensive reviews covering ethnomedicinal uses, phytochemical contents, and scientifically proven pharmacological properties of leaves, shoots, flowers, stems, and roots of the species [13].

### 3.2. One-Factor-at-a-Time (OFAT) Technique

The literature has it that experimental parameters can be narrowed down to determine the most significant cause of an important factor using the one-factor-at-a-time (OFAT) technique. The technique monitors one parameter at a time, while maintaining status quo for other parameters. Response surface methodology (RSM) was used to perform the optimization procedure. A similar approach was used by [14,15] in their optimization studies. It has also been cited by [16] that the OFAT technique had assisted screening of suboptimal conditions before performing an optimization procedure.

The OFAT technique utilized in the present optimization study proved to be helpful in finding the range for each factor more effectively. The ranges obtained as previously presented in Table 1 were used as the minimum and maximum values for each factor in the BBD design using RSM to estimate the optimum conditions for the extraction of antioxidants with the highest yield.

### 3.3. Optimization of Antioxidant Activities Using a Response Surface Methodology (RSM)

The importance of antioxidants in preventing disease has been highlighted due to their ability to inhibit free radical activities that impact human health [17]. Antioxidant phytochemicals, which are found in a wide variety of foods and medicinal plants, are important for both prevention and treatment of diseases brought on by oxidative stress. These antioxidants possess strong anti-inflammatory, antioxidant, and free radical scavenging properties, which have been cited to act as building blocks for additional bioactivities and health benefits such as anti-aging, anti-cancer, and protective effects against common and chronic diseases [18]. In an attempt to standardize the extraction of flavonoids as an important group of plant bioactive compounds, the extraction parameters in the present study were optimized using BBD coupled with RSM using Design Expert software. Free radical scavenging activity (DPPH), total flavonoid content (TFC), and total phenolic content (TPC) of *M. malabathricum* leaf extracts were examined using four independent variables including extraction temperature (X_ET_), ultrasonic time (X_UT_), solvent concentration (X_SC_), and sample-to-liquid ratio (X_SLR_) where the significance of each value and variable had been tested using the OFAT technique.

The present study suggests that RSM was the appropriate statistical analysis to establish the optimum antioxidant responses while avoiding a waste of resources. RSM has also been cited to be an effective statistical method for predicting the interaction between measured response parameters and a range of experimental variables which are expected to have an impact on the responses [18].

#### 3.3.1. Model Fitting

The results of the present study showed responses of antioxidant activity in terms of the percentage of inhibition of DPPH, the amount of TPC found, and the amount of TFC found under 29 conditions as suggested by the software presented in Table 4. Data were collected for analysis of the coefficients of the second-order polynomial equation.

Box–Behnken design is more proficient and most powerful than other designs such as the three-level full factorial design and central composite design (CCD) despite its poor coverage of the corner of nonlinear design space [19].

Data obtained from the experiments were recorded as experimental (exp.) values. The predicted (pred.) values present the values of the variables predicted based on the regression analysis calculated by the software, whose significance was to determine the residual values in regression analysis. Each actual value had a predicted value, and hence each data point had one residual. The residuals played a vital role to validate the obtained regression model. Residuals are represented graphically by means of a residual plot. The data points on the residual plot are spread around the horizontal axis, indicating the appropriateness of a linear regression model.

According to Table 4, antioxidant activities of DPPH inhibition, TPC, and TFC of the leaf extracts of *M. malabathricum* ranged from 75.89 to 96.356%, 312.384 to 646.211 mg GAE/g, and 76.33 to 104.68 mg QE/g, respectively. Under the optimum conditions, the experimental values of DPPH inhibition, TPC, and TFC were in agreement with the predicted values as suggested by RSM, indicating the suitability of the employment of the selected models.

In evaluating the interaction between linearity and regression coefficients in the response variables, the regression equations which accounted for variability in the response variables were analysed as suggested by [19]. The analyses of variance (ANOVA) are presented in Table 5.

Both multiple regression analysis and analysis of variance (ANOVA) were used to evaluate the effectiveness and fitness of the developed models as well as their significance. The developed models were appropriate for demonstrating the relationship between variables, as they were highly significant (*p* < 0.0001). All regression coefficients were recorded as significant except for the regression coefficients of X_UT_, X_ET_ X_SC_, X_UT_ X_SLR_, X_ET_
^2^ and X_SLR_
^2^ for DPPH, X_ET_ X_UT_, X_ET_ X_SLR_ and X_SC_ X_SLR_ for TPC, and X_ET_ and X_SC_ X_SLR_ for TFC.

The R-squared values for DPPH, TPC, and TFC were 0.9617, 0.9473 and 0.9475, respectively, whereas adj. R-squared values for DPPH, TPC, and TFC were 0.9233, 0.8946, and 0.8950 respectively. The model’s adequacy to precisely predict the experimental data was demonstrated by an increase in the values of R-squared and adj. R-squared, which were close to one [20]. The predicted R-squared value for responses of DPPH, TPC, and TFC were 0.8254, 0.7143 and 0.7477, respectively.

According to [21], the difference between the predicted R-squared and adjusted R-squared should not be greater than 0.2. Low coefficients of variation, ranging from 1.72 to 7.2, were recorded in the present study. All adequate precision values were higher than 4, indicating that the suggested prototype was a perfect model and could be used to develop the design space and to recommend the optimum conditions [22]. From the sequential model sum of squares, the highest-order polynomials were used to designate the models wherever the additional coefficient estimates were consequential, and therefore the models were not aliased [23]. Hence, for all four independent variables and responses, a quadratic polynomial model was set and fit well, following the recommendation of the software. The regression equations obtained for the independent and dependent variables for Y_1_ (DPPH), Y_2_ (TPC), and Y_3_ (TFC) are presented in Table 6.

#### 3.3.2. Response Surface Analysis (RSA) of 2,2-Diphenyl-1-Picrylhydrazyl (DPPH) Free Radical Scavenging Ability

RSA was selected to determine the best extraction parameters because of its consistent results in measuring antioxidant activity [24]. In the present study, RSA for DPPH scavenging ability was recorded to range from 75.89 to 96.36% inhibition. The ratio for maximum to minimum was 1.27. The mean value of the responses was 85.70%. The model’s F-value of 25.09 implies that the model was significant. There was only a 0.01% chance that an F-value this large could occur due to noise. The value of Prob > F of less than 0.0001 suggests that the model terms were significant. Free radical scavenging activity was significantly influenced at (*p* < 0.001) by three of four linear variables (X_ET_, X_SC_, X_SLR_), interaction parameters (X_ET_X_UT_, X_ET_X_SLR_, X_UT_X_SLR_, X_SC_X_SLR_), and only (X_SC_^2^) for quadratic parameters as previously shown in Table 4. Values greater than 0.0500 indicate the model terms were not significant. A “lack of fit F-value” of 0.99 implies that it is significant. There was a 55.18% chance that a “lack of fit F-value” this large could occur due to noise. Three of four independent variables, X_ET_, X_SC_ and X_SLR_, provided highly significant effects on enhancement of DPPH extraction. Greater response values were recorded, as there were interactions between the variables. The interactions are shown in Figure 3, which presents three dimensional (3D) plots of the interaction effects of the independent variables (X_ET_, X_UT_, X_SC_ and X_SLR_) on the yield of DPPH.

In each panel, two variables are shown to have an impact on the DPPH extraction. Figure 3a shows interaction between extraction temperature (X_ET_) and ultrasonic time (X_UT_) where opposite responses were recorded when ultrasonic time was increased and extraction temperature decreased, and vice versa. This could be due to the high temperature (55–60 °C), which enhanced the solvent’s diffusion through cell walls and amplified the effects of sonic cavitation [25]. Similar results were also reported by [26], who indicated that, when temperature rose, solvent viscidity decreased and molecule mobility accelerated. It was reported that the release of bioactive chemicals from plant cells was facilitated by raising the temperature of the extraction process. However, several thermosensitive chemicals could be degraded when the temperature exceeded 60 °C. In other words, as the temperature rose to 60 °C, the yield of flavonoids increased and was maintained at a high level but decreased below 55–60 °C when exposed for a long time, which could be related to the denaturing of some heat-sensitive chemicals.

In the present study, there was no significant interaction between extraction temperature (X_ET_) and solvent concentration (X_SC_) (Figure 3b). However, at low temperatures, there was an interaction between extraction temperature (X_ET_) and sample-to-liquid ratio (X_SLR_) (Figure 3c). Figure 3d reveals the important correlation between ultrasonic time (X_UT_) and solvent concentration (X_SC_). Water and ethanol, which are polar protic solvents, have been cited to be able to stabilize phenol homologues and lessen the nucleophiles’ reactivity.

Due to their variations in polarity, water and ethanol have been frequently suggested for extract preparation [26]. After considering the importance of handling security and health, a binary ethanol and water solvent extraction was chosen [27]. The present finding was consistent with other studies, which suggested that a binary solvent system was preferable to a mono-solvent system (water or pure ethanol) in the extraction of phenolic compounds due to its relative polarity. Both the chemical structure of plant tissue and the solvent system’s polarity have been suggested to affect how efficiently phenolic compounds can be dissolved.

#### 3.3.3. Response Surface Analysis (RSA) of Total Phenolic Content (TPC)

The highest total phenolic content (TPC) of *Melastoma malabathricum* leaf extracts from this study was 577.03 mg GAE/g, which shows that this extraction procedure was able to enhance the extraction of TPC from this plant as compared to other extraction methods used by [28], which found 199.10 mg GAE/g of TPC, and [29], which found 292.5 mg/GAE of TPC, in their studies. Extraction temperature is a variable that promotes extraction of antioxidant compounds by enhancing the diffusion coefficient and solubility of the compounds. The RSA (Table 5) demonstrates a high regression coefficient (R^2^  =  0.9473), and the equation for Y_2_ in Table 6 shows the relationship between TPC and extraction temperature, ultrasonic time, solvent concentration, and sample-to-liquid ratio. The 3D response surface plots with the interaction effects of the four parameters on TPC are shown in Figure 4.

Extraction temperature and ultrasonic time (Figure 4a) had no significant interaction effect on the enhancement of TPC. A non-significant interaction effect was recorded between extraction time and sample-to-liquid ratio (Figure 4c). A similar effect was observed in sub-Figure 4f, in the interaction between solvent concentration (X_SC_) and the sample-to-liquid ratio. It can be observed that TPC increased linearly with the increase in extraction temperature as the solvent concentration was increased. The interaction between ultrasonic time and solvent concentration (Figure 4b), ultrasonic time and solvent concentration (Figure 4d), and ultrasonic time and sample-to-liquid ratio (Figure 4e) increased the values of TPC. According to [30], higher solubility and diffusion coefficients of polyphenols have been recorded with increased temperature, allowing for higher extraction rates. Nevertheless, it was reported that an upper limit of temperature must be adhered to in order to prevent decomposition of thermo-sensitive compounds in some flavonoids [31]. Increasing the temperature favors extraction and enhances both the solubility of solutes and the diffusion coefficient, but beyond certain extended thresholds, compound stability could be affected due to chemical and enzymatic degradation or losses by thermal decomposition.

#### 3.3.4. Response Surface Analysis (RSA) of Total Flavonoid Content (TFC)

The present study recorded RSA for TFC to range from 76.33 to 104.68 mg QE/g (Figure 5). The responses were comparatively higher for TFC value when compared to other studies on extraction of *M. malabathricum.* Comparative studies by [32,33] using conventional Soxhlet extraction (CSE), ultrasound-assisted extraction (UAE) and modified ultrasound-assisted extraction (MUAE) reported extractions yields of 40.31, 64.94, and 54.97 mg QE/g, respectively.

From the ANOVA table (Table 3), the F-value of 18.04 suggests that the model was significant. Values of prob > F were less than 0.0100, indicating that the model terms were highly significant. Values smaller than 0.1000 indicated that the model terms were highly significant, suggesting that all interactions occurred on all four parameters. X_ET_, X_UT_, X_SC_, and X_SLR_ had a highly significant interaction effect on the enhancement of TFC, except on interactions between X_SC_ and X_SLR_, which were not significant. The lack-of-fit F-value of 1.36 implies non-significance relative to pure error. Non-significance in lack of fit was considered good, as fitting the model was preferred. Adequate precision measures the signal-to-noise ratio.

The study noted that extraction temperatures (X_ET_) had no significant quadratic effects at (*p* < 0.01). However, there was a highly significant interaction effect between extraction temperatures (X_ET_) with other variables. Model graphs shown in Figure 5 provide a visual representation of the effect of each interaction. There was a good interaction effect, as shown in Figure 5a, between extraction temperature (X_ET_) and ultrasonic time (X_UT_) at (*p* < 0.001) on the formation of flavonoids. This suggests that the higher the temperature, the lower the time in enhancing production of flavonoids. Similar findings were reported by [34], who recorded that the optimum amount of total flavonoids was attained at 50–60 °C and subsequently declined at higher extraction temperatures.

The interaction effect between extraction temperature (X_ET_) and solvent concentration (X_SC_) in sub-Figure 5b reveals significantly high TFC enhancement. TFC increased linearly with an increase in extraction temperature and an increase in solvent concentration, and showed a similar trend with enhancement of TPC. The interactions between extraction temperature (X_ET_) and solvent concentration (X_SC_), between ultrasonic time (X_UT_) and solvent concentration (X_SC_), and between ultrasonic time (X_UT_) and sample-to-liquid ratio (X_SLR_) on the enhancement of TFC are shown in Figure 5a–f.

Polyphenols are a vast and assorted class of compounds, a considerable amount of which occurrs naturally in food and plants. Flavonoids are the largest and best considered class of polyphenols. Plant polyphenols are being effectively created and sold as either supplements or nutraceutical products. In spite of the fact that these compounds play an obscure role in non-supplements, a considerable amount have high beneficial properties including antioxidant, anti-mutagenic, anti-cancer, and anti-inflammatory properties that may be advantageous in preventing diseases [35].

#### 3.3.5. Verification of Predictive Model

Three independent and dependent variables for DPPH (Y_1_), TPC (Y_2_), and TFC (Y_3_) were represented under their respective optimal extraction temperatures (X_ET_), ultrasonic times (X_UT_), solvent concentrations (X_SC_), and sample-to-liquid ratios (X_SLR_) well within the experimental range. Table 6 presents the confirmation showing comparison of results predicted by the model against the outcome of a confirmation experiment. The optimum conditions recorded in the present study were 32 °C X_ET_ for 16 min X_UT_, dissolving in 70% ethanolic solvent by 1:10 X_SLR_, yielding values for antioxidant activities of 96% inhibition, 803.456 mg QE/g, and 102.972 mg GAE/g for DPPH (Y_1_), TPC (Y_2_), and TFC (Y_3_), respectively. The table shows that the experimental values were close to the predicted values as per regression models with a range of coefficient variations between 1.55 and 4.07%.

The present study finds that this was an effective design in reducing variability in the experiment. The interaction between the variables revealed that the four variables were vital procedures in producing high antioxidant activities. Hence, working at the optimum conditions was able to successfully enhance extraction of the antioxidant compounds and save cost and time.

### 3.4. Characterization of Chemical Composition of M. malabathricum Leaf Extracts

#### 3.4.1. Gas Chromatography–Mass Spectrometry (GC–MS) Analysis

Gas chromatography–mass spectrometry (GC–MS) analysis of *M*. *malabathricum* leaf extracts revealed the presence of 14 major bioactive compounds as shown in Figure 6 and Table 7. The presence of screened compounds ranged between 0.2% (6-Acetyl-beta-d-mannose and E-7-Octadecene) and 31.7% (squalene). From the results of the GC–MS spectra, the occurrence of squalene (31.7%), lactic acid (22.4%), neophytadiene (22.2%), cyclotrisiloxane hexamethyl (12.1%), and 3,7,11,15-tetramethyl-2-hexadecen-1-ol (7.4%) were the most abundant.

Similar bioactive compounds identified in the study have been cited by a number of researchers to play crucial roles in skin wound healing. For example, squalene, found most abundantly in the present optimized extract, has been reported to possess important innate immune cells involved in the development of the wound healing process [36]. Several studies also reported its ability to prevent atherosclerotic lesions [37], skin problems [38], and cancer [39]. Lactic acid (LA), identified as another major compound present in the extract, is essentially a short-chain fatty acid, such as butyric acid and propionic acid, produced as a metabolite of lactic acid bacteria, including periodontopathic bacteria [40]. These short-chain fatty acids have been cited to have positive effects on human health. Lactic acid (as sodium lactate) is a well-known part of the skin’s natural moisturizing complex and is an excellent moisturizer [41]. Neophytadiene is a potent antimicrobial and anti-inflammatory compound. [42]. It has been reported for its role as an antifungal and antioxidant [43]. Antimicrobials are a crucial component of the wound healing process, as infections brought on by various bacteria may obstruct the healing process and cause healing of wound to be delayed or even inhibited [44]. Cyclotrisiloxane hexamethyl, also present in the extract, is said to possess the same role as has been reported by [45,46].

Compound 3,7,11,15-tetramethyl-2-hexadecen-1-ol, also known as phytol [47], is widely used as a precursor for synthetic forms of vitamin E and vitamin K1, which support the immune system’s ability to fight off viruses and bacteria. It widens blood arteries to prevent blood clotting and aids in the formation of red blood cells. It facilitates the body’s absorption of vitamin K. Vitamin E is also used by cells to facilitate interactions with one another [48].

#### 3.4.2. Quadrupole Time-of-Flight Mass Spectrometry (QTOF-MS) Analysis

UHPLC-QTOF-MS/MS was used for metabolite profiling of the optimized UAE of *M*. *malabathricum* leaf extracts. The analysis was used to support results from GC–MS analysis (Table 6) and to assess its potential as a source of natural antioxidants and plant-product-based bioactive molecules. A full chromatogram is presented in Figure 7 showing the identity of the phenolic profiles confirmed using mass fragmentation analysis and mass. The mass spectrum, together with the structure of the compounds, were identified to be flavonoids. The compounds, with their typical fragments’ mass-to-charge ratios (*m*/*z*), are presented in Table 8, showing total ion count chromatographs of the phenolic compounds detected in negative electrospray ionization (ESI) mode.

The top 18 of the compounds with a relatively high concentration detected from QTOF-MS analysis were evaluated based on the highest retention time values as shown in Table 8. Compounds found in this study were identified based on a comparison of their analytical data (retention times and high-resolution mass spectra) with those of several reference standards. Compounds were unambiguously identified as prosapogenin 5 (julibroside A1), meso-inositol, macrostemonoside D, calycanthoside, castalagin, gallic acid, bistortaside, gemin D, geraniin, potentillin, Curculigo saponin K, jangomolide, isopimpinellin, quercetin, and kaempferol-3-O-β-D-glucopyranoside. Other compounds found from the extract were (25R)-26-O-β-D-glucopyranosyl-5β-furost-20(22)-en-3β,26-diol-3-O-[β-D-glucopyranosyl-(1 → 2)]-β-D-glucopyranoside, and 3,8,9-trihydeoxy-6H-benzo[c]chromen-6-one. The fragmentation patterns and pathways of the standards helped further in confirming the structures of the derivatives of the reference compounds. Compounds without reference standards were identified by determining the elemental compositions of the precursor and product ions.

The identification of the 18 phenolic compounds from the base peak chromatogram (BPC) confirmed the high medicinal value of optimized UAE of *M*. *malabathricum* leaf extracts. The literature has it that the compound (25R)-26-O-β-D-glucopyranosyl-5β-furost-20(22)-en-3β,26-diol-3-O-[β-D-glucopyranosyl-(1 → 2)]-β-D-glucopyranoside is an antioxidant. This flavonoid compound helps in reduction of the risk of chronic diseases, including cancer, coronary heart disease, and diabetes [49]. Other flavonoid compounds present were 3,8,9-trihydeoxy-6H-benzo[c]chromen-6-one. It is known that this compound is a plant metabolite that has a role as an antioxidant, a chelator, a radiation protective agent, and an antibacterial agent. It is a polyphenol and a biflavonoid [50].

Quercetin is a flavonoid that has antioxidant and anti-inflammatory effects that might help reduce swelling, kill cancer cells, control blood sugar, and help prevent heart disease [51]. Kaempferol-3-O-β-D-glucopyranoside (astragalin, AS), a major flavonoid that exists in various plants, exerts antioxidant, antitumor, anti-human immunodeficiency virus (HIV), and anti-inflammatory effects [52].

It has been reported that prosapogenin 5 (julibroside A1) possesses anti-angiogenic and anti-tumor activities [53]. Top-of-form, bottom-of-form meso-inositol helps balance certain chemicals in the body to help with mental conditions such as panic disorder, depression, and obsessive-compulsive disorder. It might also help insulin to function better [54]. These antidiabetic properties are also seen in macrostemonoside D, a recently identified fat cell-secreted factor, visfatin, which is insulin-mimetic and plays a positive role in attenuating insulin resistance and diabetes. Curculigo saponin K, also known as saponin, decreases blood lipids, lowers cancer risks, and lowers blood glucose response. A high saponin diet has been reported to have been used in the inhibition of dental caries and platelet aggregation, in the treatment of hypercalciuria in humans, and as an antidote against acute lead poisoning [55].

Several compounds found in this study also have anti-cancer properties as reported by many researchers. The compound calycanthoside is a coumarin-related compound commonly used in the treatment of prostate cancer, renal cell carcinoma, and leukemia, and also has the ability to counteract the side effects caused by radiotherapy [56]. Both natural and synthetic coumarin derivatives have drawn much attention due to their photochemotherapy and therapeutic applications in cancer as parent compounds in anticoagulant agents [56]. Potentillin belongs to a class of organic compounds known as hydrolyzable tannins. Hydrolysable tannins (HTs) are an important group of secondary plant metabolites that includes simple gallic acid derivatives, gallotannins (GTs), and elligitannins (ETs). HTs exhibit anti-cancer, anti-angiogenic, antioxidant, anti-inflammatory, and anti-ulcerative properties [57]. Jangomolide, as the steroidal lactone withaferin A (WFA), is a dietary phytochemical. It exhibits a wide range of biological properties, including immunomodulatory, anti-inflammatory, antistress, and anti-cancer activities [58]. Myo-inositol is a phytic acid that is beneficial to human health. It has been reported to have potential health benefits including a reduction in digestion of starch (which is especially beneficial to diabetics), reduction in blood cholesterol (and as a result a reduction in cardiovascular disease), prevention of kidney stones, removal of lead and other heavy metal ions, and anti-cancer activity [59].

Other compounds present in the extract have been reported to have several health-promoting effects such as gallic acid (also known as 3,4,5-trihydroxybenzoic acid), which is a trihydroxybenzoic acid classified as a phenolic acid [60]. A new tannin-related compound named bistortaside A (1) is a class of astringent polyphrenolic that binds to and precipitates proteins and various other organic compounds, including amino acids and alkaloids [61]. Gemin D (GD) is an ellagitannin found in several plant species rich in phenolic compounds. Its many beneficial properties include antioxidant and antitumoral properties [62]. Geraniin is known for its significant antioxidant activity in vitro [63]. Isopimpinellin is a primary metabolite, which are metabolically or physiologically essential metabolites said to be directly involved in an organism’s growth, development, or reproduction [64]. Castalagin is an ellagitannin, a type of hydrolyzable tannin. Castalagin and other related ellagitanins have been reported to polymerize or form complexes with anthyocyanins and flavonoids [65].

## 4. Summary and Conclusions

The present study studied modeling and optimization in the extraction of antioxidants from *M. malabathricum.* RSM coupled with BBD based on OFAT was initially applied to optimize the extraction conditions using UAE. The optimal conditions were established at 37 kHz, X_ET_ 32 °C for X_UT_ 16 min and dissolved in X_SC_ 70% ethanol concentration by a X_SLR_ 1:10 ratio. Under these conditions, the optimum yield of DPPH inhibition and TPC were 96% ± 1.48 inhibition, 803.456 ± 32.48 mg GAE/g, and 102.972 ± 2.51 mg QE /g, respectively. The values were in agreement with those predicted by RSM models, confirming suitability of the model employed and the success of RSM for optimization of the extraction conditions.

Phytochemical screening of the extract recorded the presence of various phytoconstituents as shown by the positive reactions with their respective test reagents and contained a good amount of major phytocompounds, the presence of which may be responsible for the numerous pharmacological activities in the species as claimed by various authors. The species has demonstrated significant phenolic and flavonoid content, suggesting potent antioxidant activities by virtue of its TPC, TFC, and radical scavenging activities (DPPH). The study has identified the species’ promising potential that could be harnessed towards the development of new therapies.

This study was intended to contribute significantly to existing empirical literature and to offer insight for further research on the medicinal significance of *M. melabathricum*. The data may provide a comparative evaluation with other studies on the species and may be used for validating or substantiating of the species’ much acclaimed medicinal use.

## Figures and Tables

**Figure 1 molecules-28-00487-f001:**
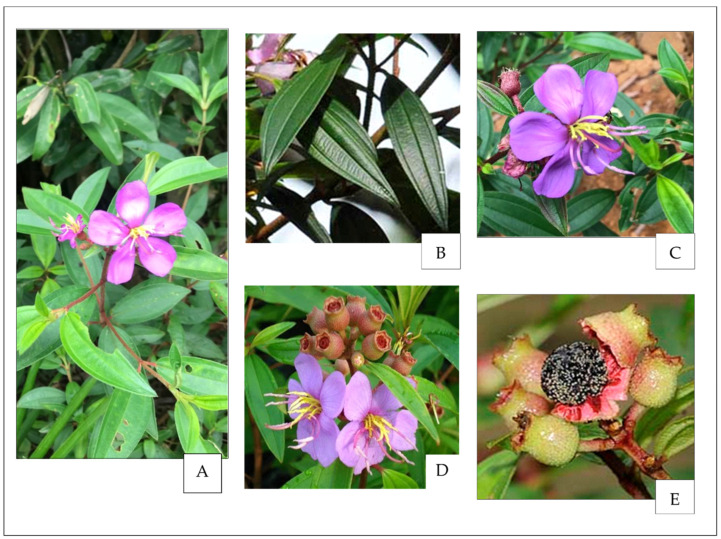
*Melastoma malabathricum* Linn. showing (**A**) matured plant, (**B**) leaves, (**C**) flower, (**D**) flowers with fruits, (**E**) seeds.

**Figure 2 molecules-28-00487-f002:**
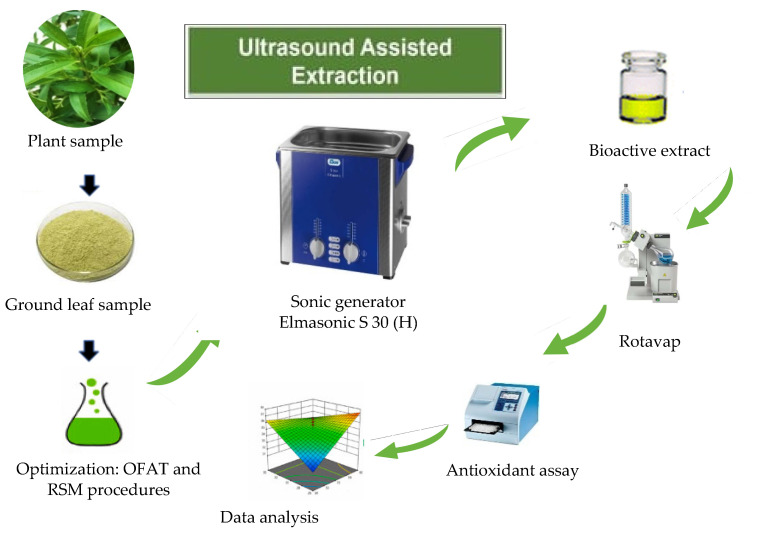
Experimental procedures in ultrasound-assisted extraction for *Melastoma malabathricum*.

**Figure 3 molecules-28-00487-f003:**
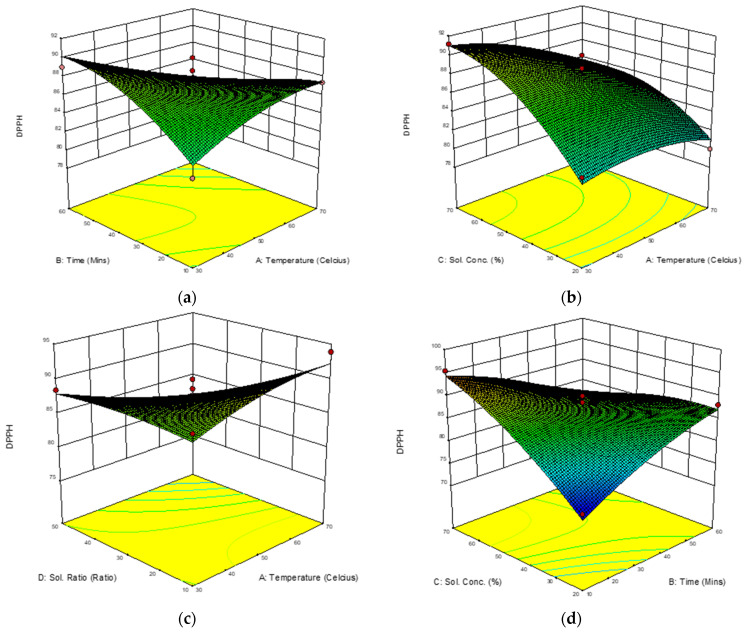
(**a**–**f**): Response surface 3D plots showing interaction effects of extraction temperatures (X_ET_), ultrasonic times (X_UT_), solvent concentration (X_SC_), and sample-to-liquid ratios (X_SLR_) on percentage of DPPH inhibition.

**Figure 4 molecules-28-00487-f004:**
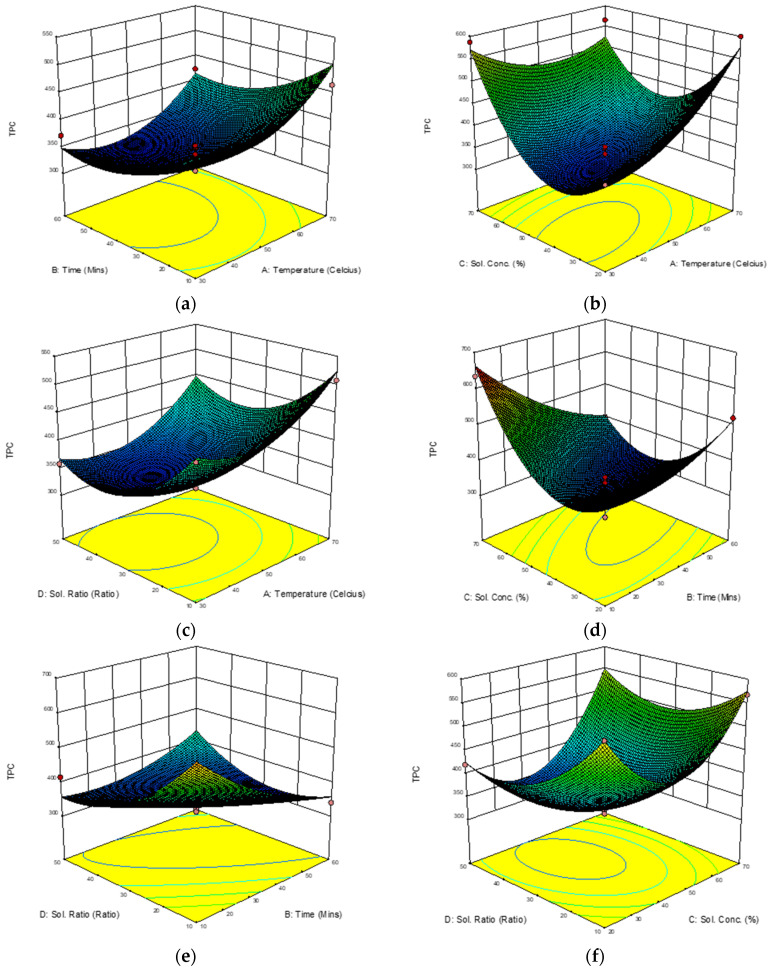
(**a**–**f**): Response surface 3D plots showing the interaction effects of extraction temperatures (X_ET_), ultrasonic times (X_UT_), solvent concentrations (X_SC_), and sample-to-liquid ratios (X_SLR_) on enhancement of TPC in mg GAE/g.

**Figure 5 molecules-28-00487-f005:**
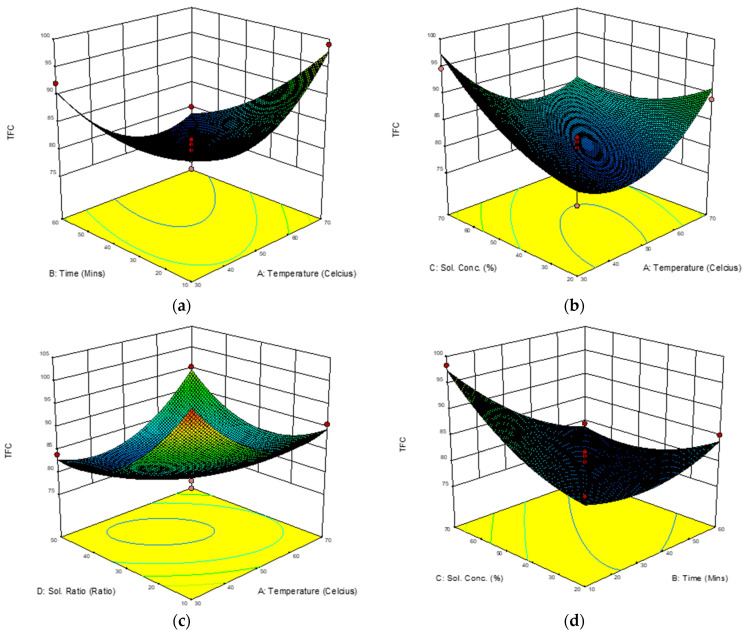
(**a**–**f**): Response surface 3D plots showing the interaction effects of extraction temperatures (X_ET_), ultrasonic times (X_UT_), solvent concentration (X_SC_), and sample-to-liquid ratios (X_SLR_) on enhancement of TFC in mg QE/g.

**Figure 6 molecules-28-00487-f006:**
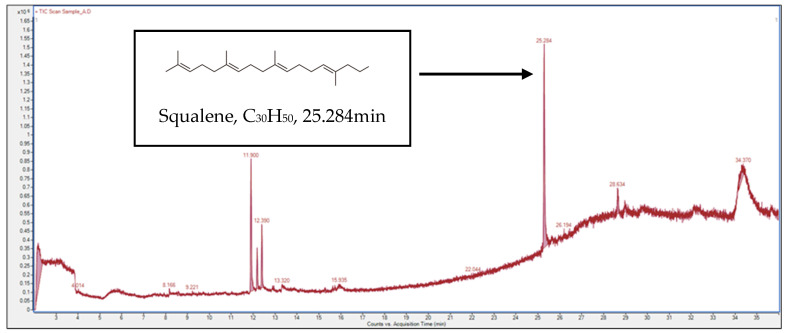
Chromatogram profile of ultrasonic-assisted extraction of *M*. *malabathricum* leaf extract using GC–MS in negative ion mode.

**Figure 7 molecules-28-00487-f007:**
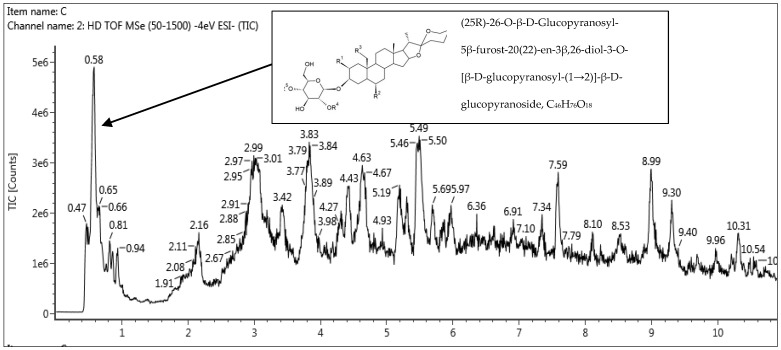
QTOF-MS chromatogram profile of optimized UAE of *M. malabathricum* leaf extracts in negative.

**Table 1 molecules-28-00487-t001:** Minimum and maximum levels of four factors in terms of coded and uncoded values.

	Minimum Level	Maximum Level
	−1	1
Extraction temperature, X_ET_	30	70
Ultrasonic time, X_UT_	10	60
Solvent concentration, X_SC_	20	70
Sample-to-liquid ratio, X_SLR_	10	50

**Table 2 molecules-28-00487-t002:** Optimization conditions for antioxidant extraction using Box–Behnken design.

Run Order ^a^	Extraction Temperature, °C	Ultrasonic Time, min	Solvent Concentration, %	Sample-to-Liquid Ratio
1	50	35	45	1:30
2	50	60	20	1:30
3	30	35	70	1:30
4	50	10	45	1:10
5	50	60	45	1:50
6	30	35	45	1:10
7	50	10	70	1:30
8	30	60	45	1:30
9	50	10	20	1:30
10	50	35	20	1:10
11	70	35	20	1:30
12	50	35	70	1:50
13	50	35	45	1:30
14	50	35	20	1:50
15	50	60	70	1:30
16	30	35	45	1:50
17	50	35	70	1:10
18	50	35	45	1:30
19	70	60	45	1:30
20	50	35	45	1:30
21	70	35	45	1:50
22	70	35	70	1:30
23	50	10	45	1:50
24	50	60	45	1:10
25	50	35	45	1:30
26	30	35	20	1:30
27	30	10	45	1:30
28	70	10	45	1:30
29	70	35	45	1:10

^a^ Randomized.

**Table 3 molecules-28-00487-t003:** Operating conditions employed in UHPLC system.

Power:	
Capillary Voltage	1.50 kV
Reference Capillary Voltage	3.00 kV
Cone Flow Rate (L/H):	
Source Temperature	120 °C
Desolvation Gas Temperature	550 °C
Desolvation Gas Flow	800 L/H
Cone Gas Flow	50 L/H

**Table 4 molecules-28-00487-t004:** Experimental (exp.) and predicted (pred.) values for antioxidant activities of inhibition of DPPH, TFC, and TPC under suggested extraction conditions.

Run Order ^a^	DPPH ^b^		TPC ^c^		TFC ^d^	
	Exp.	Pred.	Exp.	Pred.	Exp.	Pred.
1	90.00	87.89	352.31	334.09	81.00	79.43
2	88.10	87.11	519.99	515.92	85.10	83.92
3	91.17	90.99	587.43	570.74	94.66	97.32
4	88.99	89.06	646.21	595.68	100.43	102.98
5	80.09	80.93	421.00	432.24	85.01	87.18
6	88.88	87.80	449.80	457.74	104.68	103.24
7	95.44	94.32	634.40	663.91	98.36	97.43
8	89.00	90.15	371.16	347.57	92.11	90.63
9	75.89	74.71	399.90	419.68	82.99	81.54
10	78.99	80.56	564.14	564.47	90.88	89.91
11	79.99	81.08	599.63	577.03	88.99	91.04
12	80.00	79.64	535.50	549.01	89.26	87.62
13	86.88	87.89	348.80	334.09	76.33	79.43
14	83.00	83.15	420.00	425.95	84.33	83.33
15	81.78	80.84	400.00	405.67	79.36	78.70
16	88.43	87.92	357.77	367.63	83.97	82.92
17	96.36	97.41	567.50	575.39	97.88	96.28
18	86.25	87.89	321.30	334.09	79.87	79.43
19	79.94	79.90	423.29	415.26	80.00	78.63
20	87.68	87.89	335.67	334.09	78.06	79.43
21	78.00	76.96	431.05	448.56	95.34	94.67
22	83.66	85.20	565.25	525.35	83.87	86.04
23	82.00	82.92	415.53	357.45	80.99	83.19
24	89.99	89.98	340.11	358.90	80.11	82.63
25	88.64	87.89	312.38	334.09	81.88	79.43
26	82.41	81.78	384.47	385.08	79.11	81.65
27	82.44	83.69	401.10	422.98	89.00	87.76
28	87.38	87.44	464.40	501.84	98.99	97.86
29	93.88	92.27	507.77	523.35	90.66	89.60

^a^ Run order: Randomized; ^b^ DPPH: 2,2-diphenyl-1-picrylhydrazyl radical scavenging ability (% inhibition); ^c^ TPC: total flavonoid content (mg gallic acid equivalent (GAE)/g); ^d^ TFC: total phenolic content (mg quercetin equivalent (QE)/g).

**Table 5 molecules-28-00487-t005:** Analysis of variance (ANOVA) for 2,2-diphenyl-1-picrylhydrazyl free radical scavenging ability (DPPH), total phenolic content (TPC), and total flavonoid content (TFC) by surface quadratic model.

Variance Source	df	DPPH				TPC				TFC			
		Sum of Squares	Mean Square	F-Value		Sum of Squares	Mean Square	F-Value		Sum of Squares	Mean Square	F-Value	
Model	14	765.99	54.71	25.09	**	2.66 × 10^5^	18,978.35	17.98	**	1597.5	114.11	18.04	**
Extraction temp., X_ET_	1	31.62	31.62	14.50	**	16,108.06	16,108.06	15.26	**	2.69	2.69	0.43	ns
Ultrasonic time, X_UT_	1	0.87	0.87	0.40	ns	19,681.54	19,681.54	18.65	**	200.63	200.63	31.72	**
Solvent conc., X_SC_	1	133.51	133.51	61.22	**	13,463.4	13,463.4	12.76	**	85.24	85.24	13.48	**
Sample-to-liquid ratio, X_SLR_	1	173.02	173.02	79.34	**	20,392.5	20,392.5	19.32	**	174.3	174.3	27.56	**
X_ET_ X_UT_	1	49.00	49.00	22.47	**	31.22	31.22	0.03	ns	122.14	122.14	19.31	**
X_ET_ X_SC_	1	6.48	6.48	2.97	ns	14,082.36	14,082.36	13.34	**	106.82	106.82	16.89	**
X_ET_ X_SLR_	1	59.52	59.52	27.29	**	58.63	58.63	0.056	ns	161.18	161.18	25.48	**
X_UT_ X_SC_	1	167.31	167.31	76.73	**	31,415.77	31,415.77	29.77	**	111.42	111.42	17.62	**
X_UT_ X_SLR_	1	2.12	2.12	0.97	ns	24,269.44	24,269.44	23	**	148.12	148.12	23.42	**
X_SC_ X_SLR_	1	103.69	103.69	47.55	**	3143.87	3143.87	2.98	ns	1.07	1.07	0.17	ns
X_ET_ ^2^	1	7.01	7.01	3.21	ns	16,565.59	16,565.59	15.7	**	270.17	270.17	42.72	**
X_UT_ ^2^	1	15.72	15.72	7.21	*	9015.33	9015.33	8.54	**	52.24	52.24	8.26	**
X_SC_ ^2^	1	28.30	28.30	12.98	**	1.10 × 10^5^	1.10 × 10^5^	103.74	**	63.58	63.58	10.05	**
X_SLR_ ^2^	1	2.42	2.42	1.11	ns	27,146.35	27,146.35	25.72	**	293.38	293.38	46.39	**
Residual	14	30.53	2.18			14,775.56	1055.4			88.54	6.32		
Lack of Fit	10	21.76	2.18	0.99	ns	13,589.96	1359	4.59	ns	68.4	6.84	1.36	ns
Pure Error	4	8.77	2.19			1185.6	296.4			20.14	5.04		
Total	28	796.52				2.81 × 10^5^				1686.04			
R-Squared		0.9617				0.9473				0.9475			
Adj. R-Squared		0.9233				0.8946				0.8950			
Pred. R-Squared		0.8254				0.7143				0.7477			
Adeq. Precision		21.377				14.116				13.606			
C.V. %		1.72				7.2				2.88			
PRESS		139.05				80,130.68				425.45			

* and **: significant at *p* ≤ 0.05 and 0.01, respectively; ^ns^ non-significant difference at *p* ≥ 0.05; C.V.: coefficient of variations; PRESS: predicted residual sum of squares for the model.

**Table 6 molecules-28-00487-t006:** Regression equations for dependent and independent variables for DPPH, TFC, and TPC.

Variable		Equation	No.
Y_1_ (DPPH)	=	87.89 – 1.62A – 0.27B + 3.34C – 3.80D – 3.50AB – 1.27AC – 3.86AD – 6.47BC – 0.73BD – 5.09CD – 1.04A^2^ – 1.56B^2^ – 2.09C^2^ – 0.61D^2^	(4)
Y_2_ (TPC)	=	334.09A + 36.64A – 40.50B + 33.50C – 41.22D – 2.79AB – 59.33AC + 3.83AD – 88.62BC + 77.89BD + 28.04CD + 50.54A^2^ + 37.28B^2^ + 129.92C^2^ + 64.69D^2^	(5)
Y_3_ (TFC)	=	79.43 – 0.47A – 4.09B + 2.67C – 3.81D – 5.53AB – 5.17AC + 6.35AD – 5.28BC + 6.09BD – 0.52CD + 6.45A^2^ + 2.84B^2^ + 3.13C^2^ + 6.73D^2^	(6)

DPPH: 2,2-diphenyl-1-picrylhydrazyl free radical scavenging ability; TFC: total flavonoid content; TPC: total phenolic content.

**Table 7 molecules-28-00487-t007:** Compounds identified in ultrasonic-assisted extraction of *M. malabathricum* leaf extract by GC-MS.

No.	Name	Formula	RT, min	*m*/*z*, %
1	Squalene	C_30_H_50_	25.284	31.7%
2	Lactic acid	C_3_H_6_O_3_	2.183	22.4%
3	Neophytadiene	C_20_H_38_	11.900	22.2%
4	Cyclotrisiloxane, hexamethyl-	C_6_H_18_O_3_Si_3_	22.044	12.1%
5	3,7,11,15-Tetramethyl-2-hexadecen-1-ol	C_20_H4_0_O	12.390	7.4%
6	Cyclohexane,1,1’-(2-propyl-1,3-propanediyl) bis-	C_18_H_34_	15.935	0.8%
7	3-Chloropropionic acid, octadecyl ester	C_21_H_41_ClO_2_	8.566	0.7%
8	1-Octadecyne	C_18_H_34_	15.291	0.7%
9	Pentanoic acid, 5-hydroxy-, 2,4-di-t-butylphenyl esters	C_19_H_30_O_3_	8.166	0.5%
10	11,13-Dimethyl-12-tetradecen-1-ol acetate	C_18_H_34_O_2_	15.727	0.4%
11	1H-Indene, 5-butyl-6-hexyloctahydro-	C_19_H_36_	12.945	0.3%
12	Oleic Acid	C_18_H_34_O_2_	13.320	0.3%
13	6-Acetyl-beta-d-mannose	C_8_H_14_O_7_	4.014	0.2%
14	E-7-Octadecene	C_18_H_36_	9.221	0.2%

RT: Retention time (min); *m*/*z*: mass-to-charge ratio.

**Table 8 molecules-28-00487-t008:** Compounds identified in the optimized ultrasonic-assisted extraction of *M. malabathricum* leaves using UHPLC-QTOF-MS/MS.

No.	Observed	Component Name	Formula	Neutral Mass (Da)	Observed (*m*/*z*)	Mass Error (ppm)
	RT (min)					
1	0.53	(25R)-26-O-β-D-Glucopyranosyl-5β-furost-20(22)-en-3β,26-diol-3-O-[β-D glucopyranosyl -(1→2)]-β-D-glucopyranoside	C_46_H_76_O_18_	916.50317	915.4956	−0.4
2	0.53	Prosapogenin 5 (Julibroside A1)	C_53_H_84_O_22_	1072.54542	1071.5399	1.7
3	0.53	Meso-inositol	C_6_H_12_O_6_	180.06339	179.0557	−2.2
4	0.55	Macrostemonoside D	C_53_H_86_O_24_	1106.5509	1105.5392	−4
5	0.55	Calycanthoside	C_17_H_20_O_10_	384.10565	383.0994	2.8
6	3.84	Castalagin	C_41_H_26_O_26_	934.07123	933.066	2.2
7	5.68	Gallic acid	C_7_H_6_O_5_	170.02152	169.0139	−2.2
8	5.68	Bistortaside	C_22_H_24_O_14_	512.11661	511.1083	−2
9	5.68	Gemin D	C_27_H_22_O_18_	634.08061	633.074	1
10	5.94	Geraniin	C_41_H_28_O_27_	952.0818	951.071	−3.7
11	5.94	Potentillin	C_41_H_28_O_26_	936.08688	935.0821	2.7
12	5.94	Curculigo saponin K	C_48_H_82_O_19_	962.54503	961.5371	−0.7
13	7.55	Jangomolide	C_26_H_28_O_8_	468.17842	467.1691	−4.4
14	10.27	Isopimpinellin	C_13_H_10_O_5_	246.05282	245.045	−2.2
15	10.27	3,8,9-Trihydeoxy-6H-benzo[c]chromen-6-one	C_13_H_8_O_5_	244.03717	243.0291	−3.3
16	10.27	Quercetin_1	C_15_H_10_O_7_	302.04265	301.0343	v3.5
17	10.27	Kaempferol-3-O-β-D-glucopyranoside	C_21_H_20_O_11_	448.10056	447.0936	0.7
18	10.27	Munjistin	C_15_H_8_O_6_	284.03209	283.0253	1.8

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
