# Peer review of "Ultrasound-Assisted Extraction of Antioxidants from Melastoma malabathricum Linn.: Modeling and Optimization Using Box–Behnken Design"

_molecules, 2023, doi:10.3390/molecules28020487_

Round 1

Reviewer 1 Report

This study assessed an optimization of ultrasound assisted extraction of Melastoma malabathricum in order to evaluate its phytochemical properties. Firstly, one factor-at-a-time (OFAT) procedure was used to select the experimental domine of potential experimental variables and finally, response surface methodology (RSM) was used to optimize the extraction conditions. The article is generally well written and will serve as a valuable resource for the survey of Melastoma malabathricum. Nevertheless, the manuscript should be improved.

Comments:

Lines 40-62. The introduction section should be more extensive.

Please include an explanation about the difference between the present manuscript and the manuscript published by Azahar et al. (2020)( https://doi.org/10.3390/app10197002). The authors shoul be also included this reference. Moreover, why did the authors choose a Box-Behnken design? Please include the reason.

Lines 195-225. Phytochemical characterization should be deeper improved.

Why did the authors choose GC and LC MS techniques for such characterization? Please include the reasons.

Include the experimental procedure to detect the bioactive compounds by GC-MS (Section 2.2.4.1). It is mandatory to include the derivation step when you use GC-MS.

Did the authors use the optimized experimental conditions by RSM? I cannot find it in the experimental description in lines 206-212. Please include it.

How many samples did you extracted?

Lines 371-397. Please include in the discussion Table 4.

Lines 400-465. Why have the authors found only a minimum in the RSM with TPC and TFC? Are the authors sure that the experimental domine is correct? Explain it.

Reviewer 2 Report

The manuscript Ultrasound Assisted Extraction of Antioxidants from Melastoma 

malabathricum Linn.: Modeling and Optimization Using Box-Behnken Design, describes some interesting results, however, it needs a deep revision.

Comments

Methods:

Line 163: DPPH radical assay is NOT a parameter to establish antioxidant activity, at least you must include another radical scavenging method and compare them with a lipidic model, after that, you could propose your extracts as a potential antioxidant.

Some references to review: J. Agric. Food Chem. 2015, 63, 40, 8765–8776; J. Agric. Food Chem. 2014, 62, 19, 4251–4260

Line 185: units of absorbance in nm?

Line 200: international units of meters is “m” not “Mts”!!! 

Line 201: capillary column with 0.25 m of film thickness?

Line 202: how do you inject 1 L in a capillary column?

Line 228: please revised de units for MS scan. 

Results and discussion

Lines 131-133: How much extract did you use? What was the yield of the US-assisted extraction? That is, what was the initial weight of plant material used in each trial and the final yield of extracts recovered?

Lines 237-255: These paragraphs are missing. Nothing to report about your results or discussion!!! Please be more direct and succinct.

Lines 258-265: Re-write, you have to start with the results obtained, this paragraph could be in the introduction.

Lines 268-272: this could be move to methods. Here you must show the results!!!

Line 280: please re-write, because as it is written it seems that percentage will be for DPPH, TPC and TFC.

Line 293: in methods you stated: The content of flavonoids in the extracts was expressed in terms of quercetin equivalent, QE (mg of quercetin/g of extract), however here, you change the units to "mM QE/g", what is the reason?

Table 4: the percentage inhibition is not an appropriate unit to represent radical scavenging activity, as it depends on the concentration of the sample or compounds, instead the IC50 should be used for any comparison.

Lines 377, 381: what is temperature “rose” and viscidity?

Figure 3: units for DPPH?

Line 409: TPC is not produced by extraction methods or temperature, they are extracted!!!!

Figure 4: units for TPC?

Figure 5: units for TFC?

Lines 488-491: what do you mean with “ranged betweee… X%? percentage of? How did you quantify these compounds?

Figure 6: Change the title, it is either a chromatogram profile or a mass spectrum, but not both. According to LCMS or GCMS coupled techniques, the obtained graphic is named Total Ion Current (TIC).

Table 7: what do you mean when write: m/z %??? For example, 31.7 % m/z??? Please revise it.

Table 8: what is the deviation of the m/z values observed? could you show the fragmentation patterns, at least for the most abundant ones?

All references have to be in the same format.

 In summary, in addition to addressing the suggestions raised above, with all due respect, I recommend that the authors make more effort to improve the presentation of the results in a clearer manner and to provide more rationale for the discussion. Therefore, as it stands, I recommend that it be thoroughly improved to be considered for publication in this journal.

Reviewer 3 Report

The manuscript “Ultrasound Assisted Extraction of Antioxidants from Melastoma malabathricum Linn.: Modeling and Optimization Using Box-Behnken Design" is devoted to the modeling and optimization of ultrasound assisted extraction (UAE) of Melastoma malabathricum. Free radical scavenging, total phenolic content and total flavonoid content were estimated for obtained extracts. TWIMS-QTOFMS, gas chromatography-mass spectrometry were applied for phytocompounds identification. Quite a large number of methods of characterization of the obtained extract have been applied. It is obvious that the authors comprehensively characterize the extracts obtained and optimize the extraction method according to various indicators. To solve optimization problems, the authors successfully apply a combination of One-factor-at-a-time and Response surface methodology coupled with Box-Behnken design of experiment. The experimental results are very well described by approximation equations, which is demonstrated by high values of R^2. 14 components were identified by GC-MS, and 19 – by TWIMS-QTOFMS. It is important data from the phytochemistry point of view. The work makes a good impression, the data is sufficient, well described. The study is logically constructed and has a completed form.

The results obtained can be useful from the point of view of biochemistry and phytochemistry.

I think, this manuscript can be published in the Molecules in the present form.

Author Response

Thank you for the responses and comments given for the manuscript submitted

Round 2

Reviewer 2 Report

The authors have satisfactorily answered most of the questions, however,  The question "what was the deviation from the observed m/z values?" was not answered. Please provide an appropriate response.
